The most complete enantiornithine from North America and a phylogenetic analysis of the Avisauridae

http://orcid.org/0000-0003-1081-0552 Atterholt Jessie 1 2 3 jessie.atterholt@gmail.com
Hutchison J. Howard 1
http://orcid.org/0000-0002-3898-8283 O’Connor Jingmai K. 4 5
1 Department of Integrative Biology, University of California , Berkeley, CA , USA
2 Raymond M. Alf Museum of Paleontology , Claremont, CA , USA
3 Graduate College of Biomedical Sciences, Western University of Health Sciences , Pomona, CA , USA
4 Key Laboratory of Vertebrate Evolution, Institute of Vertebrate Paleontology and Paleoanthropology, Chinese Academy of Sciences , Beijing , People’s Republic of China
5 CAS Center for Excellence in Life and Paleoenvironment , Beijing , People’s Republic of China
Knoll Fabien
Electronic publication date: 2018 Nov 13
Publication date: 2018
Volume: 6
Electronic Location ID: e5910
Received 2018 Mar 28; Accepted 2018 Oct 8
Copyright: © 2018 Atterholt et al.
Copyright year: 2018
Copyright holder: Atterholt et al.
License: This is an open access article distributed under the terms of the Creative Commons Attribution License, which permits unrestricted use, distribution, reproduction and adaptation in any medium and for any purpose provided that it is properly attributed. For attribution, the original author(s), title, publication source (PeerJ) and either DOI or URL of the article must be cited.
License URL: https://creativecommons.org/licenses/by/4.0/

Keywords: Enantiornithes, Evolution, Late Cretaceous, Fossil bird

Funding: Annie M. Alexander endowment to the UCMP National Natural Science Foundation of China grant number 41688103 Howard Hutchison’s field work was supported by the Annie M. Alexander endowment to the UCMP. Jingmai O’Connor’s research was supported by the National Natural Science Foundation of China (grant number 41688103). The funders had no role in study design, data collection and analysis, decision to publish, or preparation of the manuscript.

==============================
The most complete known North American enantiornithine was collected in 1992 but never formally described. The so-called “Kaiparowits avisaurid” remains one of the most exceptional Late Cretaceous enantiornithine fossils. We recognize this specimen as a new taxon, Mirarce eatoni (gen. et sp. nov.), and provide a complete anatomical description. We maintain that the specimen is referable to the Avisauridae, a clade previously only known in North America from isolated tarsometatarsi. Information from this specimen helps to clarify evolutionary trends within the Enantiornithes. Its large body size supports previously observed trends toward larger body mass in the Late Cretaceous. However, trends toward increased fusion of compound elements across the clade as a whole are weak compared to the Ornithuromorpha. The new specimen reveals for the first time the presence of remige papillae in the enantiornithines, indicating this feature was evolved in parallel to dromaeosaurids and derived ornithuromorphs. Although morphology of the pygostyle and (to a lesser degree) the coracoid and manus appear to remain fairly static during the 65 million years plus of enantiornithine evolution, by the end of the Mesozoic at least some enantiornithine birds had evolved several features convergent with the Neornithes including a deeply keeled sternum, a narrow furcula with a short hypocleidium, and ulnar quill knobs—all features that indicate refinement of the flight apparatus and increased aerial abilities. We conduct the first cladistic analysis to include all purported avisuarid enantiornithines, recovering an Avisauridae consisting of a dichotomy between North and South American taxa. Based on morphological observations and supported by cladistic analysis, we demonstrate Avisaurus to be paraphyletic and erect a new genus for “A. gloriae,” Gettyia gen. nov.

Introduction

The Enantiornithes are a diverse group of Cretaceous land birds first recognized by Walker (1981) from an assemblage of isolated, three-dimensionally preserved bones collected from deposits of the Maastrichtian Lecho Formation at the El Brete locality in Argentina (Chiappe, 1993, 1996; Walker & Dyke, 2009). The disarticulated and isolated nature of the “El Brete” material left open the possibility that the Enantiornithes was paraphyletic (Steadman, 1983). However, through the discovery of an articulated partial skeleton later used to erect the taxon Neuquenornis volans, Chiappe (1992) demonstrated both the validity of the Enantiornithes as well as the avian affinity of an enigmatic clade, the Avisauridae. Today, the Enantiornithes are considered the first major avian radiation and the dominant clade of land birds in the Cretaceous. Their remains have been collected on every continent except Antarctica, in some cases occurring in great abundance, and by the Late Cretaceous they appear to have occupied a wide range of ecological niches including one potentially flightless form (O’Connor, Chiappe & Bell, 2011).

Most data regarding the Enantiornithes come from the 125–120 Ma Jehol Group where thousands of nearly complete and fully articulated specimens have been uncovered (Zhou & Zhang, 2007) accounting for approximately half the currently recognized diversity. Although nearly complete, these specimens are typically two-dimensionally preserved, and elements are often split between two slabs such that fine anatomical details often cannot be discerned. In contrast, relatively few specimens have been collected from Upper Cretaceous deposits. Only four taxa are represented by partial skeletons: the holotype of Parvavis chuxiongensis (Wang, Zhou & Xu, 2014), the only reported Late Cretaceous bird from China; the potentially flightless Elsornis keni from Mongolia (Chiappe et al., 2007); Neuquenornis volans from Argentina (Chiappe & Calvo, 1994); and Nanantius valifanovi from Mongolia, here considered a junior synonym of Gobipteryx minuta (Elzanowski, 1974; Kurochkin, 1985; Chiappe, Norell & Clark, 2001). An additional taxon, “Gobipipus” (Kurochkin, Chatterjee & Mikhailov, 2013) was erected based on fairly complete embryonic remains, but “Gobipipus” and other juvenile specimens (Elzanowski, 1981) lack diagnostic features and are not useful for comparison with the adult material studied here.

The remainder of the Late Cretaceous record is far more incomplete, consisting of small associations of fragmentary elements and individual isolated, partial elements, many used to erect new taxa. These are primarily collected in South America but are also known from North America, Madagascar, and Eurasia (e.g., Yungavolucris brevipedalis, Soroavisaurus australis, Bauxitornis mindszentyae, Flexomornis howei, Incolornis martini) (Chiappe, 1993; Dyke & Ősi, 2010; O’Connor & Forster, 2010; Tykoski & Fiorillo, 2010; Panteleev, 2018). Although a majority of Late Cretaceous taxa are based on single complete—or even fragmentary—elements, the material tends to be well preserved in three-dimensions revealing anatomical details that are rarely preserved in two-dimensional specimens from the Early Cretaceous. This makes comparison between enantiornithines early and late in their evolution difficult and hinders the study of evolutionary trajectories within the clade (O’Connor, 2009).

The enantiornithine fossil record is particularly poor in North America and entirely limited to the Late Cretaceous (Fig. 1). The first probable enantiornithines from North America were collected in the 19th Century and consisted of three metatarsal fragments found in the Wyoming Lance Formation, including an incomplete metatarsal III that may be referable to A. archibaldi (Chiappe & Walker, 2002). Since then, several more unnamed fragments have been described including RAM 14306, a partial coracoid also from the Kaiparowits Formation (Farke & Patel, 2012), a distal tibiotarsus from Dinosaur National Monument (Buffetaut, 2010), and several fragmentary coracoids from the Hell Creek Formation (Longrich, Tokaryk & Field, 2011). Slightly more complete specimens were used to erect taxa. In Montana, two taxa were named from isolated tarsometatarsi: Avisaurus archibaldi from the Hell Creek Formation (Brett-Surman & Paul, 1985) and Avisaurus gloriae from the Two Medicine Formation (Varricchio & Chiappe, 1995). Alexornis antecedens from the Bocana Roja Formation in Baja California (Brodkorb, 1976), Halimornis thompsoni from the Mooreville Chalk Formation of Alabama (Chiappe, Lamb & Ericson, 2002), and Flexomornis howei from the Woodbine Formation of Texas (Tykoski & Fiorillo, 2010) are all known from small associations of fragmentary elements. The most complete of these, Halimornis, consists only of a proximal humerus, partial scapula, distal femur, pygostyle, and a thoracic vertebra (Chiappe, Lamb & Ericson, 2002). A proximal humerus from New Mexico has been referred to the genus Martinavis, which otherwise occurs in Argentina and France (Walker, Buffetaut & Dyke, 2007). However, the first probable enantiornithines collected in North America were three metatarsal fragments found in the Lance Formation in Wyoming during the 19th Century, including an incomplete metatarsal III that may be referable to A. archibaldi.

Figure 1 Map of the enantiornithine fossil record in North America.

Empty black circle—individual incomplete element. Filled black circle—individual complete element. Filled red circle—associated fragments and elements (<10% of skeleton preserved). Yellow star—associated fragments and elements (>10% of skeleton preserved).

Originally, A. archibaldi was described as a member of a new clade of non-avian theropod dinosaurs, the Avisauridae (Brett-Surman & Paul, 1985). An isolated tarsometatarsus belonging to the original El Brete collection described by Walker (1981) was also referred to the clade (Brett-Surman & Paul, 1985) and later used to erect a new taxon, Soroavisaurus australis (Chiappe, 1993). With the new data provided by the discovery of Neuquenornis, Chiappe (1992) provided support for the enantiornithine affinity of the Avisauridae, consisting of these three taxa. Subsequently, several additional taxa have been referred to this clade (including A. gloriae; Varricchio & Chiappe, 1995), some through phylogenetic analysis (e.g., Enantiophoenix) and others based only on morphological observations (e.g., Bauxitornis, Intiornis) (Cau & Arduini, 2008; Dyke & Ősi, 2010; Novas, Agnolín & Scanferla, 2010).

Here, we describe the full anatomy of UCMP 139500, the most complete enantiornithine collected in North America to date. Although originally found in 1992, it has only been preliminarily described in an abstract, referring the specimen to the Avisauridae (Hutchison, 1993). Like most other Late Cretaceous enantiornithines, UCMP 139500 is well-preserved in three-dimensions and contributes substantially to our understanding of Late Cretaceous skeletal anatomy as a whole, as well as with regard to the North American Avisauridae, otherwise known only from isolated tarsometatarsi. The specimen was found disarticulated weathering out of a clay rip-up clast in a paleo-channel deposit. We provide a complete anatomical description of the “Kaiparowits avisaurid,” re-assess the validity of the Avisauridae through the first phylogenetic analysis to include all purported members of the clade, and discuss the impact of UCMP 139500 on existing hypotheses regarding evolutionary trends within the Enantiornithes.

Methods

Systematic paleontology

Class AVES Linnaeus, 1758

ORNITHOTHORACES Chiappe, 1995

Subclass ENANTIORNITHES Walker, 1981

Family AVISAURIDAE Brett-Surman and Paul, 1985

Revised diagnosis: Enantiornithine birds with the following unique combination of morphological features: tarsometatarsus with inclined proximal articular surface; strong transverse convexity of the dorsal surface of the mid-shaft of metatarsal III; a distinct plantar projection of the medial rim of the trochlea of metatarsal III (unambiguously supported in our phylogenetic analysis); and a laterally compressed J-shaped metatarsal I (modified from Chiappe (1993)).

Phylogenetic definition: the last common ancestor of Neuquenornis volans and Avisaurus archibaldi plus all its descendants (Chiappe, 1993).

Included genera: Avisaurus (Brett-Surman & Paul, 1985); Soroavisaurus (Chiappe, 1993); Neuquenornis (Chiappe & Calvo, 1994); Intiornis (Novas, Agnolín & Scanferla, 2010); Mirarce (current study); and Gettyia (current study).

MIRARCE GEN. NOV.

Etymology: Named for its spectacular preservation and level of morphological detail (Latin “mirus” for wonderful), and after Arce, winged messenger of the titans in Greek mythology, for the evidence suggesting a refined flight apparatus in this species.

Diagnosis: As for the type and only known species, given below.

Type species: Mirarce eatoni sp. nov. (by monotypy)

Etymology: The type species is named in honor of Dr. Jeffrey Eaton, for his decades of work contributing to our understanding of the Kaiparowits Formation and the fossils recovered from it.

MIRARCE EATONI SP. NOV

Holotype: UCMP 139500, a three-dimensional partial skeleton consisting of several cervical and thoracic vertebrae (including the axis), the pygostyle, almost all phalanges from the left pes and several from the right, a complete humerus, femur, and tarsometatarsus, a partial scapula, coracoid, furcula, and tibiotarsus, as well as fragments of the sternum, radius, ulna, carpometacarpus, and manual phalanges (see Table 1 for measurements of select elements).

Table 1 Measurements of skeletal elements of Mirarce eatoni.

Measurement	Mirarce eatoni	
Axis, length	18.1	
Pygostyle, length	37.9	
Coracoid, width of sternal end	27.0	
Furcula, length	∼65.9	
Furcula, width of omal tip (avrg.)	7.7	
Furcula, width of hypocleidium	2.5	
Scapula, length	>63.2	
Scapula, width	8.3	
Humerus, length	95.9	
Humerus, width	11.6	
Ulna, width	∼10.7	
Femur, length	89.0	
Femur, width	8.4	
Tarsometatarsus, length	48.1	
Tarsometatarsus, width	10.9	
Note:

“Length” refers to proximodistal/craniocaudal length; “width” refers to the width of the midshaft/midpoint of the element unless otherwise specified. All measurements are given in units of mm.

Type horizon and locality: UCMP locality V93097, Late Cretaceous (late Campanian 76–74.1 Ma; Roberts, Deino & Chan, 2005) Kaiparowits Formation of Grand Staircase-Escalante National Monument in Garfield County, Utah, USA.

The electronic version of this article in portable document format will represent a published work according to the International Commission on Zoological Nomenclature (ICZN), and hence the new names contained in the electronic version are effectively published under that Code from the electronic edition alone. This published work and the nomenclatural acts it contains have been registered in ZooBank, the online registration system for the ICZN. The ZooBank Life Science Identifiers (LSIDs) can be resolved and the associated information viewed through any standard web browser by appending the LSID to the prefix http://zoobank.org/. The LSIDs for this publication are: genus name—urn:lsid:zoobank.org:act:A90E4FD8-999B-4B7D-BE33-B288D10FC8E8; species name—urn:lsid:zoobank.org:act:58005BE1-E4F5-4B7C-9A0C-FD94A0B80F30; publication LSID—urn:lsid:zoobank.org:pub:269CEBCA-EC05-425D-B71D-1A5507C8E48B. The online version of this work is archived and available from the following digital repositories: PeerJ, PubMed Central, and CLOCKSS.

Diagnosis. A large, turkey-sized avisaurid (see above diagnosis) enantiornithine (thoracic vertebrae with centrally located parapophyses; pygostyle cranially forked with ventrolateral processes; furcula dorsolaterally excavated; Chiappe & Walker, 2002) with the following autapomorphies: posterior end of sternum weakly flexed caudodorsally, terminating in a small knob; ulnae with remige papillae present; small, deep, circular pit located just craniolateral to the femoral posterior trochanter; small, triangular muscle scar on medial margin of the femoral shaft just distal to the head followed distally by a much larger proximodistally elongate oval; distinct, rugose ridge-like muscle attachment located on the craniomedial margin of the femur a quarter length from the distal end; and tubercle for the m. tibialis cranialis located at the mid-point of the shaft of metatarsal II on the dorsal surface. The new species is further distinguished by the unique combination of the following characters: acrocoracoidal tubercle very weakly developed and medially located; furcula with truncate (untapered) omal tips weakly developed into articular facets and oriented perpendicular to the axis of the rami; ventral projection of the sternal keel proportionately greater than in most other enantiornithines (similar to condition observed in Neuquenornis); acetabulum fully perforate; medial surface of the medial condyle of the tibiotarsus with deep circular excavation; and elongate, slightly raised, flat, oval surface present on the medial edge of the plantar surface of metatarsal II continuous with a weak medial plantar crest.

Differential diagnosis. Compared to other avisaurids: metatarsals entirely unfused except for the proximal ends (proximal 1/3 fused in A. archibaldi; III and IV fused distally in “A. gloriae”); proximal articular surface very weakly inclined (slightly less than in A. archibaldi; strongly inclined in “A. gloriae”); trochlea of metatarsal IV weakly lunate in plantar view (strongly lunate in A. archibaldi); tubercle for the m. tibialis cranialis developed on the dorsomedial surface of metatarsal II is more distally located than in A. archibaldi (slightly less distal than in “A. gloriae”); elongate, slightly raised, flat, oval surface on medial plantar surface of metatarsal II more elongate and proximally located in A. archibaldi; and asymmetry of condyles in the trochlea of metatarsals II and III is less developed compared to the condition in A. archibaldi.

Ontogenetic assessment. Gross morphology indicates the specimen was an adult at the time of death. All preserved compound elements (e.g., the distal tibiotarsus, distal carpometacarpus) are fused to the extent typically observed in other enantiornithines, including the tarsometatarsus, in which the distal tarsals fuse to the metatarsals relatively late in enantiornithine development (Hu & O’Connor, 2017). Although considered an adult, size may have increased and fusion continued to progress given that protracted growth is observed in other Late Cretaceous enantiornithines (Chinsamy, Chiappe & Dodson, 1995) and that other North American avisaurids show a greater degree of fusion between the metatarsals.

Description

Axial skeleton

Cervical vertebrae. Three cervical vertebrae are preserved, including the axis (Figs. 2A–2E). The peg-like dens projects dorsal to the atlantean articular facet; its craniocaudal length is 1.5 times its mediolateral width. The articular facets of the postzygapophyses are oriented ventrally with slight lateral deflection and are medially continuous with each other through a thin shelf of bone that overhangs the vertebral foramen. The epipophyses are strongly developed but do not extend caudally beyond the caudal margin of the postzygapophyses. The caudal articular surface of the axis appears weakly heterocoelic.

Figure 2 A sampling of the best-preserved cervical and thoracic vertebrae, including the axis.

(A) Axis in lateral view. (B) Axis in dorsal view. (C) Axis in caudal view. (D) Third cervical vertebra in lateral view. (E) Third cervical vertebra in ventral view. (F) Posterior cervical vertebra in lateral view. (G) Posterior cervical vertebra in ventral view. (H) Thoracic vertebra in lateral view. (I) Thoracic vertebra in ventral view. (J) Thoracic vertebra in anterior view. Abbreviations: ds, dens; ep, epipophysis; lg, lateral groove; lr, lateral ridge; pap, parapophysis; prz, prezygopophysis; poz, postzygopophysis; ps, posterior shelf; sp, spinous process; vp, ventral process. Scale bar equals one cm. Photos: David Strauss.

The two post-axial cranial cervical vertebrae are substantially longer than the axis (approximately 1.5 times its length). The prezygapophyses are flat (epipophyses absent), cranioventral-caudodorsally oriented, and sub-lachriform (tapered dorsally with a straight medial margin and a convex lateral margin). A low but even neural spine extends nearly the entire length of the centrum. The ventral surface of the centrum also appears to form a low keel. The caudal articular surface of the vertebra is dorsoventrally concave and mediolaterally convex.

Dorsal vertebrae. Two dorsal vertebrae were found; they are amphicoelous with slightly concave articular surfaces that are much larger than the vertebral foramen (Figs. 2F–2J). The vertebrae are spool-shaped with deep grooves excavating the lateral surfaces. In one vertebra, a lateral groove appears to be perforated by a foramen in the cranial portion, but this may be a preservational artifact. The ventral surface is not keeled as in Elsornis and some El Brete specimens (Chiappe et al., 2007). The parapophyses are centrally located, as in other enantiornithines (Chiappe & Walker, 2002). The spinous process is as dorsoventrally tall as the centrum, narrowest at its base, and slightly displaced caudally (not centered on the centrum), typical of enantiornithines (Chiappe & Walker, 2002). The transverse processes are not preserved.

Pygostyle. The fully fused pygostyle shows several features typically characteristic of enantiornithines (Wang et al., 2017). The proximal end bears a craniodorsal fork formed by the prezygapophyses of the first fused vertebra. In dorsal view, these processes define a deep U-shaped concavity (Fig. 3), whereas they form a V-shaped incisure in Halimornis. The cranial fork is continuous with the dorsolateral margins. The dorsal surface is gently concave and wider than the ventral surface. Ventrally the pygostyle bears a prominent pair of laminar ventrolateral processes, which extend 80% the length of the pygostyle and taper distally without the pronounced constriction present in some taxa (e.g., Halimornis, Longipteryx; Zhang et al., 2001) or medial invagination present in the caudal margin of the pygostyle in pengornithids (Wang et al., 2017).

Figure 3 Pygostyle.

(A) Dorsal view. (B) Ventral view. (C) Left lateral view. (D) Right lateral view. (E) Cranial view. Abbreviations: df, dorsal fork; dlp, dorsolateral processes; mf, median furrow; vlp, ventrolateral processes. Scale bar equals one cm. Photos: David Strauss. Illustrations: Gregory C. Arena.

Thoracic girdle

Furcula. The furcula is nearly-complete and well preserved, with only moderate mediolateral crushing (Fig. 4). It is Y-shaped with an interclavicular angle of approximately 40°, less than observed in many Early Cretaceous taxa (Wang et al., 2014). The omal halves of the clavicular rami are subparallel, whereas they typically are more widely splayed in Early Cretaceous taxa. Although the narrow interclavicular angle in this specimen may be somewhat exaggerated by crushing, this morphology appears comparable with the South American avisaurid Neuquenornis. Another similarity between these two taxa is a short hypocleidium (though we note that this structure may be incomplete in Neuquenornis), measuring less than 1/4 the length of the clavicular rami in Mirarce, compared to half the length or more in many Early Cretaceous enantiornithines (Wang et al., 2014). The omal tips of the furcular rami are weakly expanded before they abruptly truncate. The omal margin is concave, presumably forming a facet for articulation with the coracoid, and oriented perpendicular to the long axis of the rami.

Figure 4 Furcula.

(A) Dorsal view. (B) Ventral view. Abbreviations: dg, dorsal groove; hyk, hypocleidial keel; itr, intermuscular ridge. Scale bar equals one cm. Photos: David Strauss. Illustrations: Gregory C. Arena.

Sternum. Only the xiphoid process of the sternum was recovered (Fig. 5). The lateral margins are straight in dorsal and ventral view as in most enantiornithines, whereas the xiphial margin demarcates a wide V-shape (xiphoid process absent) in primitive enantiornithines (e.g., Protopteryx, Pengornithidae) (Hu, Zhou & O’Connor, 2014). The dorsal surface is weakly concave. Ventrally, the narrow process bears a well-developed keel, similar to that in preserved in Neuquenornis, that decreases in height caudally. In contrast, the keel of Elsornis and Early Cretaceous enantiornithines is poorly developed along the xiphoid process (O’Connor, 2009). The posterior end is weakly flexed caudodorsally and terminates in a small knob not observed in other enantiornithines.

Figure 5 Xiphoid process of sternum.

(A) Ventral view. (B) Left lateral view. (C) Ventral view. (D) Right lateral view. Abbreviations: tk, terminal knob; vk, ventral keel. Scale bar equals one cm. Photos: David Strauss. Illustrations: Gregory C. Arena.

Scapula. The shaft of the left scapula is present, but the proximal and the caudal extremities are not preserved (Fig. 6). The scapular blade is straight in mediolateral view. The cranial half of the costal surface is excavated by a shallow fossa defined by a thickening of the dorsal margin of the blade, a morphology also observed in Elsornis, Halimornis, and Neuquenornis (Chiappe, Lamb & Ericson, 2002; Chiappe & Dyke 2006; Chiappe & Calvo, 1994). The caudal half of the lateral surface is also excavated by a shallow, elongate fossa, as in Halimornis. Although obfuscated by breakage, the preserved portion of the scapular blade weakly tapers distally.

Figure 6 Left scapula.

(A) Medial view. (B) Lateral view. Abbreviations: mf, medial fossa. Scale bar equals one cm. Photos: David Strauss. Illustrations: Gregory C. Arena.

Coracoid. The omal half and sternal margin of the left coracoid are preserved as separate fragments. As in other enantiornithines the acrocoracoid, glenoid, and scapular cotyla are proximodistally aligned in dorsal view (Chiappe & Walker, 2002) (Fig. 7). The acrocoracoid is straight and rounded, typical of most enantiornithines (Chiappe & Walker, 2002; Panteleev, 2018), whereas this process is medially hooked in pengornithids (Hu, Zhou & O’Connor, 2014). The acrocoracoidal tubercle is weakly developed and blunt, in contrast to the angular tubercle observed in Enantiornis leali (Walker, 1981) and RAM14306 (Farke & Patel, 2012). In medial view a groove partially separates the weakly convex scapular cotyla from the glenoid, just distal to the tubercle; this feature is more strongly developed in RAM 14306. The neck is perforated medially by a supracoracoidal nerve foramen as in some enantiornithines (Chiappe & Walker, 2002; Panteleev, 2018) (Fig. 7). The dorsal surface of the neck is weakly concave; this becomes a large fossa distal to the supracoracoidal nerve foramen. Similar well-developed dorsal fossae are present in a number of Late Cretaceous enantiornithines (Chiappe & Walker, 2002; Panteleev, 2018). The sternal margin lacks a lateral process, as in other enantiornithines. The medial angle is acute, the lateral angle is caudally directed, and the sternal margin is concave, as in Enantiornis and Elsornis. The sternal margin is thickest along the medial portion, and the corpus narrows laterally, as in many other enantiornithines (Wang et al., 2015a). Although difficult to determine unequivocally because the element is incomplete, it appears the sternal margin was not angled caudolaterally, as in some Late Cretaceous enantiornithines (Buffetaut, 1998).

Figure 7 Partial left coracoid.

(A) Lateral view. (B) Medial view. (C) and (E) Dorsal view. (D) and (F) Ventral view. Abbreviations: apr, acoracoidal process; atu, acoracoidal tubercle; gf, glenoid facet; gr, medial groove; la, lateral angle; ma, medial angle; sc, scapula. Photos: David Strauss. Illustrations: Gregory C. Arena.

Thoracic limb

Humerus. The humerus is fairly short and robust. The proximal end is typically enantiornithine in profile: concave centrally rising dorsally and ventrally (whereas it is straight to convex in the basal Pengornithidae) (Chiappe & Walker, 2002; Hu et al., 2015; Zelenkov, 2017) (Fig. 8). The cranial surface of the humerus is deeply concave, a shape exaggerated by the presence of a centrally located circular fossa which is also present in most other Late Cretaceous enantiornithines (e.g., Enantiornis; Chiappe, 1996, Gurilynia; Kurochkin, 1999; Martinavis Walker, Buffetaut & Dyke, 2007), and which may represent the coracobrachial impression (O’Connor, 2009). The deltopectoral crest is 0.4 times the length of the humerus, proportionately longer than in other Late Cretaceous taxa, and is nearly half the width of the shaft. The distal end truncates rapidly such that the crest is rectangular, typical of enantiornithines (O’Connor, 2009). The cranioventral surface of the bicipital crest is deeply excavated by a pit possibly for the m. scapulohumeralis caudalis (Chiappe & Walker, 2002). Similar to Halimornis, the enlarged, ridge-like ventral tubercle is separated by a deep capital incision, which wraps around to the proximal articular surface of the humerus and is continuous with the bicipital crest (Chiappe, Lamb & Ericson, 2002). As in other Late Cretaceous enantiornithines the ventral tubercle is excavated by a pneumotricipital fossa (this is perforated only in PVL 4022; Chiappe & Walker, 2002). A second pneumatic fossa is continuous with the capital incision just distal to the proximal articular surface.

Figure 8 Left humerus.

(A) Cranial view. (B) Caudal view. (C) Lateral view. (D) Medial view. (E) Proximal view. (F) Distal view. Abbreviations: bc, bicipetal crest; bf, brachialis fossa; cbi, coracobrachial impression; ci, capital incisure; dc, dorsal condyle; dpc, deltopectoral crest; fp, flexor process; hh, humeral head; ptf, pneumotricipital fossa; vc, ventral condyle. Scale bar equals one cm. Photos: David Strauss. Illustrations: Gregory C. Arena.

The humeral shaft is bowed. The distal end is craniocaudally compressed and mediolaterally expanded, as in other Late Cretaceous enantiornithines (e.g., Martinavis, Enantiornis) (Chiappe, 1996; Chiappe & Walker, 2002). The condyles are slightly staggered such that the dorsal condyle is proximal to the ventral condyle and a deep intercondylar incisure cuts between them. This incisure is weakly angled proximoventral-distodorsally and is parallel to the long axis of the dorsal condyle. The ventral condyle is transversely oriented, as in other enantiornithines (Chiappe & Walker, 2002). A large flexor process is present, projecting distally such that the distal margin is angled as in some other enantiornithines (e.g., Alexornis) (Chiappe & Walker, 2002). The olecranon fossa is present only as a shallow incision between the flexor process and condyles. Tricipital grooves are absent.

Ulna. The right ulna is preserved as a mineral cast of the endosteal cavity with only small fragments of cortex present. Together, these indicate that the bone was slightly bowed, as in other enantiornithines and most basal birds. Distally, the semilunate ridge (external condylar ridge) is strongly developed. Two rugose patches preserved on the caudal margin of the shaft are interpreted as quill knobs (remige papillae) (Fig. 9). These prominent rugosities are elongated in parallel to the long axis of the bone. Although breakage of the fossil makes it impossible to determine the length of each papilla, to measure the spacing between them, or to estimate the number of secondary feathers, recognition of these structures for the first time in an enantiornithine is highly significant.

Figure 9 Quill knobs on Mirarce eatoni and Pelecanus occidentalis.

(A) Right ulna of Mirarce. (B) Close-up photo of the ulna of a modern Pelecanus. Abbreviation: qk, quill knobs. Scale bar equals one cm. Photos: Dave Strauss.

The quill knobs in Mirarce are much more substantial than those reported in non-avian theropods (e.g., Velociraptor; Turner, Makovicky & Norell, 2007, Concavenator; Ortega, Escaso & Sanz, 2010), bearing a closer similarity to quill knobs seen in modern birds (Edington & Miller, 1942; Hieronymus, 2015). Morphology of remige papillae varies among extant birds to the extent that Livezey & Zusi (2007) use four states to characterize this feature in their comprehensive phylogenetic analysis of living birds. Quill knobs, when present, range from small impressions to prominent tuberculae. In some taxa with very prominent papillae, they vary from distinct, separate knobs (e.g., Platalea leucorodia, Tadorna ferruginea; Edington & Miller, 1942) to more elongate tumescences that are connected along a thin, continuous ridge (e.g., Sagittarius serpentarius, Antigone antigone, Neophron percnopterus; Edington & Miller, 1942). The quill knobs of Mirarce appear most similar to these latter taxa but are flatter and wider than observed in modern birds.

Radius. Proximal and distal ends of a right radius are preserved, along with fragments of the shaft that indicate that an interosseous groove, like that observed in some other enantiornithines (e.g., Enantiornis) was absent (Chiappe & Walker, 2002). Proximally the circular humeral cotyla is concave. The distal fragment of the right radius preserves the radiocarpal and ulnar articular surfaces oriented at a 90° angle and separated by a small, distally-projecting tubercle. The radiocarpal articular face is visible, forming a bluntly triangular facet on the dorsal surface of the radius; it covers the entire distal margin and even has a small extension on the ventral surface.

Carpometacarpus. A small fragment is identified as the carpal trochlea of the right carpometacarpus reveals asymmetry in the carpal trochlea, as in some living birds (e.g., Phalacrocorax, Lagopus, Gallus), with the dorsal condyle projecting farther (Figs. 10A–10D). The fragment strongly suggests the carpometacarpus was fully fused at the proximal end, as in all adult enantiornithines. Another fragment is interpreted as the distal end of metatarsal II; the distal articular surface for the first phalanx is heart shaped. This fragment shows no signs of even partial fusion to the minor metacarpal, as in Neuquenornis and all other known enantiornithines (Chiappe & Walker, 2002).

Figure 10 Identifiable preserved fragments of right manus.

(A) Carpometacarpus in lateral view. (B) Carpometacarpus in medial view. (C) Carpometacarpus in ventral view. (D) Carpometacarpus in proximal view. (E) First phalanx of major digit in ventral view. (F) First phalanx of major digit in dorsal view. (G) First phalanx of major digit in lateral view. (H) First phalanx of major digit in medial view. Abbreviations: cat, carpal trochlea; crp, cranial pillar; uaf, ulnocarpal articular facet; vr, ventral ridge. Scale bar equals one cm. Photos: David Strauss.

Manual phalanges. The only identifiable manual element is the first phalanx of the major digit (Figs. 10E–10H). The cranial margin is flat and wide, and the caudal margin is keeled forming a triangular cross section. As in other enantiornithines, the phalanx lacks the caudal expansion and dorsoventral compression that is present in ornithuromorphs (O’Connor, Chiappe & Bell, 2011). Breakages reveal large, pneumatic chambers in interior of the phalanx.

Pelvic girdle

Only the fused portions of the pelvic elements contributing to the left acetabulum were recovered (Fig. 11). The antitrochanter is small, triangular, and laterally oriented. It is positioned on the caudodorsal margin of the acetabulum as in other enantiornithines (Chiappe & Walker, 2002). The preserved portion of the ilium indicates a dorsal antitrochanter was present along the dorsal margin forming a tubercle-like expansion of the laterodorsal iliac crest located just over the antitrochanter, as in some other enantiornithines (e.g., Sinornis; Sereno & Chenggang, 1992; PVL4042; Chiappe & Walker, 2002). Only the ventral half of the proximal pubis is present; at least along this portion, the pubic shaft was not mediolaterally compressed. The large, circular acetabulum appears to be fully perforated, whereas it is partially occluded in the Early Cretaceous Qiliania graffini (Ji et al., 2011).

Figure 11 Pelvic girdle fragment.

(A) Lateral view. (B) Medial view. Abbreviations: ace, acetabulum; ant, antitrochanter; dat, dorsal antitrochanter; ili, ilium; isc, ischium; pub, pubis; sac, supracetabular crest. Scale bar equals one cm. Photos: David Strauss. Illustrations: Gregory C. Arena.

Pelvic limb

Femur. The right femur is complete and well preserved (Fig. 12). Fragments of the left femur were also found, comprising the proximal end (minus the head) and the medial half of the distal end. The femur is long, almost equal to the humerus in length and nearly twice the length of the tarsometatarsus. The femoral shaft is bowed cranially, as in most Early Cretaceous enantiornithines and Martinavis; this curvature is more pronounced in the distal half of the element, as in Martinavis (Chiappe & Walker, 2002; Walker & Dyke, 2009). The femoral neck is relatively distinct and elongate, similar to other Late Cretaceous enantiornithines (e.g., Martinavis), and projects slightly dorsally at a proximomedial angle (whereas it projects laterally in PVL 4037; Chiappe & Walker, 2002). A fossa for the capital ligament is not present on the femoral head (though present in femora from El Brete, for example, PVL 4060, PVL 4037), but there is a distinct flattening and rugosity where the ligament would have presumably attached. The trochanteric crest is robust and thick, although it thins caudally. It projects proximally slightly less than the femoral head and is angled craniolaterally—caudomedially. On the cranial surface, it is laterally angled to form the craniolateral margin of the femur; as it diminishes along this surface, it becomes continuous with an intermuscular line (potentially of the mm. femorotibialis lateralis and femorotibialis intermedius) (Baumel et al., 1993). This line angles medially and extends midway down the bone before splitting near the center of the cranial surface and diminishing away a quarter length from the distal margin. In two Hungarian enantiornithine femora (MTM PAL 2011.20, -.21; Ösi, 2008) this intermuscular line is not split. The lateral surface of the femur is excavated caudolaterally by a deep posterior trochanter, as in other enantiornithines. Inside the excavation lies a delicate, cranially convex, semilunate muscle scar. The excavation of the posterior trochanter forms a laterally oriented boney shelf, as in other enantiornithines. This shelf formed by the posterior trochanter is craniocaudally convex (similar to PVL-4060) and opens on the caudal surface (Chiappe & Walker, 2002). A unique feature present on both femora is a small, deep, circular pit located just craniolateral to the posterior trochanter (Fig. 12C). This may represent the point of insertion for major hip flexors, such as the mm. iliotrochantericus cranialis and medius (Mosto, Carril & Picasso, 2013).

Figure 12 Right femur.

(A) Cranial view. (B) Caudal view. (C) Medial view. (D) Lateral view. (E) Proximal view. (F) Distal view. Abbreviations: cil, cranial intermuscular line intermuscular line; lc, lateral condyle; lgt, lateral gastrocnemial tubercle; lil, lateral intermuscular line; iicm, insertion of m. iliotrochantericus cranialis and medius; mc, medial condyle; ofm, origin of m. femorotibialis medialis; pt, posterior trochanter. Scale bar equals one cm. Photos: David Strauss. Illustrations: Gregory C. Arena.

The medial margin of the shaft just distal to the head bears a small, triangular muscle scar followed distally by a much larger proximodistally elongate oval not observed in any other known enantiornithine (Fig. 12). In modern birds, the m. femorotibialis medialis has an elongate point of origin along the medial surface of the femur (Mosto, Carril & Picasso, 2013). The more prominent muscle scar present in Mirarce may possibly be analogous, suggesting a short and wide origin for the m. femorotibialis medialis.

In caudal view, three distinct intermuscular lines are present (Fig. 12B). The first originates on the medial surface near the proximal end of the large muscle scar, curving around to the caudal surface at an angle, ending in a large oval rugosity (forming the more proximal rugosity). Just distal to this landmark, another intermuscular line runs parallel to the long axis of the femoral shaft, almost on the caudomedial margin of the shaft; distally it is truncated by a distinct, rugose, ridge-like muscle attachment located a quarter length from the distal end (forming the more distal rugosity). The third line, located on the caudolateral margin of the bone, extends from the posterior trochanter distally. Near its distal end, this intermuscular line is interrupted by a small, raised, lachriform muscle scar, possibly the lateral gastrocnemial tubercle (Baumel et al., 1993). A fourth, faint intermuscular line parallels the middle section of the laterocaudal muscular line; these two intermuscular lines define a thick strip of rugose bone that extends from the proximal rugosity to the distal rugosity, likely representing a major site of muscle attachment. A lateral intermuscular line is also reported in an indeterminant enantiornithine from the Late Cretaceous of Madagascar (FMNH PA 752; O’Connor & Forster, 2010) and MTM V.2002.05 from Hungary, although in these specimens the lines do not terminate in a scar for muscle attachment.

Distally, the region between the condyles has been moderately crushed, but an intercondylar sulcus does not appear to have extended onto the cranial surface of the femur. On the caudal surface, a shallow popliteal fossa is present. A weak circular impression possibly for the origin of the cranial cruciate ligament is observed between the two condyles (Baumel et al., 1993). The medial condyle is much larger than the lateral condyle, as in living birds and other enantiornithines (Chiappe & Walker, 2002). The medial condyle is broad, rounded, and tapered medially, while the lateral condyle is more ridge-like and mediolaterally compressed. A prominent lateral epicondyle is present, as well as a small impression for the attachment of the collateral ligament, but the fibular trochlea is poorly developed as in other enantiornithines (O’Connor, 2009). A caudally-projecting lateral flange like that present in some enantiornithines (e.g., Enantiornis, Neuquenornis, Concornis) (Chiappe & Walker, 2002) is also absent, although the lateral condyle does project further caudally and distally relative to the medial condyle. The medial surface of the medial condyle bears a deep circular excavation also present in Martinavis (PVL 4036).

Tibiotarsus. Only the proximal and distal ends of the right tibiotarsus were recovered (Fig. 13). The proximal articular surface is weakly convex with a low tubercle developed in one area, similar to observations of Soroavisaurus (Chiappe, 1993). Distally the medial condyle is much larger than the lateral condyle, following the plesiomorphic state for enantiornithines (Chiappe & Walker, 2002). As in other enantiornithines, the two condyles contact, whereas they are separated by an intercondylar incisure in most ornithuromorphs. The medial and lateral surfaces of the condyles are both excavated by a deep circular pit present in some other well-preserved enantiornithines (e.g., Qiliania; Ji et al., 2011). On the lateral side, the pit is closed caudodorsally by a small tubercle, similar to PVL 4021, 4027 (Chiappe & Walker, 2002).

Figure 13 Preserved fragments of right tibiotarsus.

(A) Proximal view. (B) Lateral view. (C) Cranial views. Abbreviations: dt, distal tubercle; lc, lateral condyle; lte, lateral trochlear excavation; mc, medial condyle; pt, proximal tubercle. Scale bar equals one cm. Photos: David Strauss. Illustrations: Gregory C. Arena.

Tarsometatarsus. The left tarsometatarsus is complete (Fig. 14); the right tarsometatarsus is missing the distal portion. The tarsometatarsus is proportionately short and wide, similar to other avisaurids (Fig. 15). The proximal and distal ends of the metatarsals are approximately the same width, and show no distal expansion, as in Yungavolucris brevipedalis, or distal narrowing, as in Lectavis bretincola (Chiappe, 1993). The metatarsals are proximally fused to the distal tarsals and to each other but are otherwise unfused throughout their lengths. In proximal view, the slightly concave medial cotyla is much wider than the flatter lateral cotyla, consistent with the widths of the tibiotarsal condyles. The two cotylae are separated by a weak convexity that continues onto the surface of the lateral cotyle—like other enantiornithines an intercotylar eminence is absent. The entire proximal articular surface is weakly angled, similar to A. archibaldi but lacking the extreme tilt observed in “A. gloriae” (Fig. 15). The proximal articular surface is expanded to form a circumferential labum that overhangs the shaft of the tarsometatarsus. The labum is thickest on the plantar surface in the location of the neornithine hypotarsus, similar to the condition in basal ornithuromorphs. The plantar surface of the labum may represent the origin of the m. extensor hallucis longus, or may have supported a cartilaginous hypotarsus (Jiang et al., 2017). The proximocranial margin of the medial cotyla slopes mediodistally. The center of the proximocaudal margin is slightly concave, also as in A. archibaldi, and bears a small, dorsally-directed tubercle level with the intercotylar contact. In proximal view, the lateral cotyle tapers laterally and bears a minute, proximally-directed tubercle on the lateral margin, again also present in A. archibaldi.

Figure 14 Left tarsometatarsus.

(A) Dorsal view. (B) Plantar view. (C) Medial view. (D) Lateral. (E) Proximal view. (F) Distal view. Abbreviations: lc, lateral cotyle; mc, medial cotyle; mtIa?, metatarsal I articulation?; mtII, metatarsal II; mtIII, metatarsal III; mtIV, metatarsal IV; pl, proximal labum; std, supratrochlear depression; tct, tibialis cranialis tubercle. Scale bar equals one cm. Photos: David Strauss. Illustrations: Gregory C. Arena.

Figure 15 Comparison of avisaurid tarsometatarsi showing variation in size of the element and location of the tibialis cranialis tubercle, among other morphological variations.

(A) Avisaurus archibaldi. (B) Mirarce eatoni. (C) Gettyia gloriae. (D) Bauxitornis mindszentyae. (E) Sauroavisaurus australis. Abbreviations: tct, tibialis cranialis tubercle. Scale bar equals one cm. Illustrations: Gregory C. Arena.

As in most other enantiornithines, the metatarsals are aligned in a single dorsoplantar plane (Fig. 14). Metatarsals II and IV are straight throughout their lengths as in A. archibaldi, but differing from the more curved metatarsals of “A. gloriae.” No vascular foramina are observed. As in other avisaurids, metatarsals II–IV are subequal in width; metatarsal IV is only marginally thinner than metatarsals II and III in dorsal view, although it is much more delicate in lateral view where a dorsoplantar compression is observable (especially at the midshaft). The long axis of the cross-section is dorsomedial-lateroplantarly oriented such that the lateral margin of metatarsal IV forms a weak plantar crest, laterally defining the excavated plantar surface of the tarsometatarsus. The dorsal surface of metatarsal III is strongly convex, as in other avisaurids; the dorsal surfaces of metatarsals II and IV are nearly flat. The well-developed tubercle for attachment of the m. tibialis cranialis on metatarsal II is located nearly at the mid-point of the element, intermediate between the position in A. archibaldi and “A. gloriae.” As in all avisaurids, the tubercle is located on the dorsomedial margin of metatarsal II, whereas it is more laterally located in some enantiornithines and most ornithuromorphs (O’Connor, 2009). An elongate, slightly raised, flat, oval surface is present on the medial edge of the plantar surface of metatarsal II, continuous with a weak medial plantar crest; a similar feature is present is present in the holotype of A. archibaldi, but in this taxon is more elongate and proximally located. This feature probably represents the attachment site of a muscle, such as the m. abductor digiti II that originates on this region of metatarsal II in modern birds (Venden Berge & Zweers, 1993), or a strong ligament, such as the medial plantar ligament of the tarsometatarsus, which also occupies a similar position in modern taxa. This surface may alternatively represent the articular surface for metatarsal I, which is not demarcated in other enantiornithines (O’Connor, 2009). However, in other avisaurids (Nequenornis and Soroavisaurus) metatarsal I articulates on the medial surface of metatarsal II, and no enantiornithine preserves a distinct surface on metatarsal II to indicate the articulation with metatarsal I. This fossa in living birds is slightly concave whereas the facet in A. archibaldi and UCMP 139500 is flat. The length of the oval facet is also inconsistent with its identification as the metatarsal I fossa; metatarsal I is proportionately shorter in most enantiornithines (except the basal Pengornithidae). A similar scar is also present in some dromaeosaurids; although early interpretations considered this to be the metatarsal I fossa, this feature has since been reinterpreted as the possible origin of the digits I and II flexor and abductor tendons (Norell, Makovicky & Mongolian-American Museum Paleontological Project, 1999). Therefore, we favor interpretation of this facet as the attachment point of a ligament potentially associated with the hallucal joint.

A deep and narrow medial intertrochlear incisure extends the distal third of the tarsometatarsus, along with a much smaller lateral intertrochlear incisure (Fig. 14). In distal view, the metatarsal trochleae are nearly coplanar, with II and IV slightly angled toward metatarsal III. The distal end of metatarsal II is slightly deflected and expanded medially as in A. archibaldi and “A. gloriae.” The trochlea of metatarsal II is ginglymous; there is no collateral ligament fovea on the medial surface of the trochlea, although there is a slight tubercle in this region where the ligament could have inserted. This fovea is well developed on the lateral surface of the metatarsal II trochlea and the lateral and medial surfaces of metatarsal III, and weakly developed on the lateral surface of metatarsal IV (is absent in A. archibaldi). The trochlea of metatarsal II is subequal in width to that of metatarsal III but the distal margin is slightly angled such that the distal margin of the medial condyle is proximal to the lateral condyle. The trochlea of metatarsal III is ginglymous; the dorsal surface bears a small, pit-like dorsal trochlear depression continuous with an intercondylar sulcus, a morphology not developed in A. archibaldi. In dorsal view, the medial condyle is slightly larger than the lateral condyle. In distal view, the medial condyle projects farther plantarly and is 86% the thickness of the lateral condyle. In comparison, the medial condyle is 73% the thickness of the lateral condyle in A. archibaldi. In plantar view the medial condyle tapers sharply and extends proximally farther than the lateral condyle. The trochlea of metatarsal IV is reduced to a single condyle as in other enantiornithines (O’Connor, Averianov & Zelenkov, 2014); in dorsal view, the proximomedial margin of the trochlea bears a very small, medially-directed tubercle that suggests an enclosed vascular foramen was present between metatarsals III and IV. This indicates that the metatarsals are slightly disarticulated, and that the intertrochlear incisures may be exaggerated.

Pedal phalanges. A number of isolated phalanges are preserved representing parts of both the left and right feet; 23 of the 28 pedal phalanges were collected, including nearly the entire left foot (Figs. 16 and 17). The hallucal claw is the largest in the foot. The second digit has two phalanges that are subequal in length although the second is more delicate, followed by a large claw. The first phalanx of the third digit is the longest in the foot; the following two phalanges decrease in slightly in length, followed by a claw. The phalanges of the fourth digit are all short and robust with a small claw that appears to be more recurved than that of the other digits; the first phalanx of this digit is slightly longer than the phalanges distal to it. All non-ungual phalanges have well-developed medial and lateral fovea for the attachment of the collateral ligaments. All ungual phalanges have a deep neurovascular groove.

Figure 16 Non-ungual pedal phalanges of the left foot.

(A) Proximal view. (B) Distal view. (C) Dorsal view. (D) Medial view. Elements are identified on the left by digit number and phalanx number (D#, P#). Scale bar equals 0.5 cm. Photos: David Strauss.

Figure 17 Ungual pedal phalanges of the left foot.

(A) Proximal. (B) Lateral view. Digit number is identified on the left. Scale bar equals 0.5 cm. Photos: David Strauss.

Phylogenetic analysis

In order to test existing hypotheses regarding the phylogenetic affinity of potential avisaurid taxa, we created the first data matrix to include all such taxa: the Kaiparowits specimen, A. archibaldi, “A. gloriae,” Soroavisaurus, Neuquenornis, Intiornis, Bauxitornis, Concornis, Mystiornis, Halimornis, Gobipteryx, and Enantiophoenix (Elzanowski, 1974; Brett-Surman & Paul, 1985; Sanz & Buscalioni, 1992; Chiappe, 1993; Chiappe & Calvo, 1994; Varricchio & Chiappe, 1995; Chiappe, Lamb & Ericson, 2002; Cau & Arduini, 2008; Dyke & Ősi, 2010; Novas, Agnolín & Scanferla, 2010; Kurochkin et al., 2011; O’Connor et al., 2009). The new analysis is based on the O’Connor & Zhou (2013) matrix, modified to include an additional state in character 233 and seven additional tarsometatarsal characters mostly derived from previous avisaurid analyses (see Supplemental Information) (Chiappe, 1993; O’Connor, 2009; O’Connor, Averianov & Zelenkov, 2014). We also included revised cranial scorings for Ichthyornis based on recently published data (Field et al., 2018). The modified matrix consists of 43 taxa (26 enantiornithines, 10 ornithuromorphs) scored across 252 morphological characters, which we analyzed using TNT (Goloboff, Farris & Nixon, 2008a). Early avian evolution is extremely homoplastic (O’Connor, Chiappe & Bell, 2011; Xu, 2018) thus we utilized implied weighting (without implied weights Pygostylia was resolved as a polytomy due to the placement of Mystiornis) (Goloboff et al., 2008b); we explored k values from one to 25 (see Supplemental Information) and found that the tree stabilized at k values higher than 12. In the presented analysis we conducted a heuristic search using tree-bisection reconnection retaining the single shortest tree from every 1,000 replications with a k-value of 13. This produced six most parsimonious trees with a score of 25.1. These trees differed only in the relative placement of five enantiornithines closely related to the Avisauridae, forming a polytomy with this clade in the strict consensus tree (Consistency Index = 0.453; Retention Index = 0.650; Fig. 18).

Figure 18 A cladogram depicting the hypothetical phylogenetic position of Mirarce eatoni.

This is the strict consensus tree (Consistency Index = 0.453; Retention Index = 0.650) produced from six most parsimonious trees (score of 25.1; k-value of 13). These six trees differed only in the relative placement of the five enantiornithines most closely related to the Avisauridae, which here form a polytomy with this clade.

Enantiornithes is resolved as the sister group to the Ornithuromorpha and Sapeornis is resolved as the sister taxon to Ornithothoraces. Ichthyornis, Hesperornis, Apsaravis, and Gansus form successive out groups to Neornithes. These taxa form a dichotomy with a clade formed by Yanornis + Longicrusavis; Patagopteryx and Archaeorhynchus form successive outgroups, as the basal-most ornithuromorphs. These results are consistent with previous analyses (O’Connor & Zhou, 2013; Wang et al., 2014; Hu & O’Connor, 2017). Iberomesornis + Protopteryx are resolved in a clade as the most basal enantiornithines. The Pengornithidae (Eopengornis + Pengornis) is the outgroup to a polytomy that includes Shanweiniao, Longipteryx, Longirostravis + Rapaxavis, and all more derived enantiornithines, which includes a well resolved Avisauridae. The Avisauridae consists of a dichotomy between two clades, one formed by the Kaiparowits specimen + “A. gloriae” and A. archibaldi and the other by Neuquenornis + Intiornis and Soroavisaurus. Avisauridae forms a polytomy with Enantiophoenix, Elsornis, Eoenantiornis, Halimornis, and Mystiornis (the taxa whose positions varied in equal length trees). Shenqiornis + Eocathayornis, a clade formed by Cathayornis + Concornis and Eoalulavis, and Bauxitornis + Gobipteryx form successive outgroups to this polytomy (Fig. 18).

Discussion

First briefly mentioned in a published abstract over 20 years ago (Hutchison, 1993), UCMP139500—the “Kaiparowits avisaurid”—remains the most complete known North American enantiornithine (Figs. 19 and 20). UCMP139500 was originally described as a member of the Avisauridae, the first recognized clade of enantiornithines (Brett-Surman & Paul, 1985) and only identified Late Cretaceous clade. This group was originally recognized based on isolated tarsometatarsi from North (A. archibaldi) and South America (Soroavisaurus) but soon after a partial skeleton representing a new species, Neuquenornis volans (Chiappe & Calvo, 1994) was also referred to the clade and the first phylogenetic definition was proposed (Chiappe, 1992). The clade is defined as “the common ancestor of Neuquenornis volans and Avisaurus archibaldi plus all its descendants” and diagnosed by a strong plantar projection of the medial condyle of the metatarsal III trochlea, metatarsal III midshaft cranial surface strongly convex cranial, and a J-shaped metatarsal I (Chiappe, 1992). Although alternative phylogenetic definitions have been proposed (Cau & Arduini, 2008) the results of this study support the original definitions provided by Chiappe (1992, 1993).

Figure 19 A skeletal reconstruction of Mirarce eatoni showing preserved skeletal elements (white).

Illustration: Scott Hartman.

Figure 20 A reconstruction of living Mirarce eatoni, illustrating the large body size of this taxon.

Illustration: Brian Engh.

There has been longstanding phylogenetic support for an avisaurid clade consisting of Avisaurus, Soroavisaurus, and Neuquenornis (Chiappe, 1993; Sanz, Chiappe & Buscalioni, 1995). However, since the discovery of numerous complete taxa in China, recent phylogenetic analyses targeting Mesozoic birds as a whole typically have not included fragmentary taxa such as Avisaurus and Soroavisaurus. A few of these analyses have included the more complete Neuquenornis with Gobipteryx (Zhou & Zhang, 2006; Zhou, Zhang & Li, 2009; Hu & O’Connor, 2017). Cau & Arduini (2008) considered the Avisauridae to also include Enantiophoenix, Concornis, and Halimornis, but not Gobipteryx. This suggested that the Avisauridae was present in the Early Cretaceous and not limited to New World deposits. In the most extensive analysis of enantiornithine relationships to date, all potential avisaurids with the exception of Gobipteryx formed part of a large polytomy of derived taxa (O’Connor, 2009). More recently, Intiornis and Bauxitornis, both represented by tarsometatarsi, were also referred to the Avisauridae, but these assignments were not supported through cladistic analysis (Dyke & Ősi, 2010; Novas, Agnolín & Scanferla, 2010). Mystiornis, known only from an isolated tarsometatarsus, preserves some avisaurid-like features (such as a dorsally convex metatarsal III) and has been resolved in a clade with Avisaurus outside the Enantiornithes (Kurochkin et al., 2011). However, this specimen was not referred to the Avisauridae because of the metatarsals are fully co-ossified, a feature not present in any known enantiornithine (Kurochkin et al., 2011). Fusion is heavily affected by ontogeny, and we suggest that this conclusion is not strongly justified, especially in light of the variable amount of fusion apparent in the tarsometatarsus of other avisaurids. A previous analysis attempting to explore the phylogenetic affinity of Mystiornis with regard to the Avisauridae produced a massive polytomy (O’Connor, Averianov & Zelenkov, 2014).

Mirarce is readily identified as an avisaurid, but in order to resolve existing taxonomic issues within the Avisauridae we created the first data matrix to include all potential avisaurid taxa (Mirarce, A. archibaldi, “A. gloriae,” Soroavisaurus, Neuquenornis, Intiornis, Bauxitornis, Concornis, Mystiornis, Halimornis, Gobipteryx, and Enantiophoenix) using a version of the O’Connor & Zhou (2013) matrix modified to include seven tarsometatarsal characters designed to target avisaurid relationships (Fig. 18). The results support a monophyletic New World Avisauridae restricted to the Late Cretaceous. Avisauridae consists of a dichotomy between North (Mirarce + “A. gloriae” and A. archibaldi) and South American taxa (Neuquenornis + Intiornis and Soroavisaurus) and provides the first phylogenetic support for the inclusion of Intiornis in the Avisauridae. This clade is supported by a single unambiguous synapomorphy: metatarsal III trochlea with plantarly projecting medial condyle (249:1) (see Supplemental Information for complete list of synapomorphies). The dichotomy between North and South American taxa is notable but expected. The South American clade is characterized by the midline tapering of the tibiotarsus condyles (217:0) and the presence of a proximal vascular foramen (226:1). North American avisaurids share the presence of a hypertrophied tubercle for the m. tibialis cranialis (248:1) and a medially excavated metatarsal IV trochlea (252:2). Notably, both North and South American lineages include large and small bodied taxa. UCMP 139500 is resolved as more closely related to “A. gloriae” than to A. archibaldi, a relationship supported by the shared absence of a plantarly excavated tarsometatarsus and a tubercle for the m. tibialis cranialis that is located near the midpoint. These results indicate “A. gloriae” should not be assigned to Avisaurus. This is strongly supported by major morphological differences among these three specimens (see below). Thus, we erect a new genus for “Avisaurus” gloriae, Gettyia gen. nov.

Consistent with previously observed similarities with avisaurids, Mystiornis (Kurochkin et al., 2011) and Enantiophoenix (Cau et al., 2008) form part of a polytomy with the Avisauridae, also including Late Cretaceous taxa Elsornis and Halimornis, and Eoenantiornis from the Jehol. Currently, it would be impossible to provide unambiguous support for Halimornis or Enantiophoenix as avisaurids given that these taxa do not preserve elements critically necessary to assign specimens to this clade (namely the tarsometatarsus, which is only partially preserved in Enantiophoenix and absent in Halimornis). The geographic distribution of taxa closely related to the Avisauridae (spanning nearly every continent yielding enantiornithines) provides no clues as to the origin of this clade. Other purportedly “avisaurid-like” taxa are resolved further down the tree: Concornis is resolved with other Early Cretaceous enantiornithines, potentially closely related to Cathayornis from the Jehol and the sympatric Eoalulavis; a Bauxitornis + Gobipteryx clade forms the outgroup to the Concornis clade, ambiguously supported by two synapomorphies (249:1; 240:1), neither of which is preserved in either taxon. Although Mystiornis is resolved as an enantiornithine, this unusual specimen requires further research to better understand the apparently bizarre pattern of fusion and its true phylogenetic affinity.

Revised Systematic Paleontology

GETTYIA GEN. NOV.

Etymology: Named in honor of Mike Getty, a great friend, technician, and field paleontologist, who is dearly missed.

Diagnosis: As for the type and only known species, given below.

GETTYIA GLORIAE (Varricchio & Chiappe, 1995) new comb.

Holotype: MOR 553E/6.19.91.64, a three-dimensional tarsometatarsus missing part of metatarsal IV.

Type horizon and locality: Upper Cretaceous (Campanian) Two Medicine Formation, MOR locality TM-068, Glacier County, Montana, USA.

Diagnosis: small avisaurid enantiornithine with the following unique combination of features: dorsal surface of the tarsometatarsus strongly inclined; attachment for the m. tibialis cranialis located beyond the midpoint of the tarsometatarsus; and distal vascular foramen completely closed by metatarsal IV.

Avisaurid diversity

For taxonomic purposes, comparison between avisaurids is limited to the tarsometatarsus (Fig. 15; Table 2); this element in UCMP139500 is very similar to Avisaurus archibaldi and Gettyia gloriae (formerly A. gloriae), but differs in several features which justify the erection of a new taxon, Mirarce eatoni (gen. et sp. nov). Even within the tarsometatarsus alone, the Avisauridae display considerable variation that includes differences both obvious (e.g., position of the tubercle for the m. tibialis cranialis, proximal foramen between metatarsals III and IV) and subtle (e.g., shapes of the metatarsal trochlea). The tarsometatarsus of UCMP139500 is intermediate in size between that of Avisaurus archibaldi and Gettyia gloriae. The metatarsals of A. archibaldi are fused along the proximal fifth of their lengths; in the smaller Gettyia metatarsals III and IV are also distally fused indicating that differences in body size between avisaurids species cannot be explained by ontogeny. UCMP 139500 does not demonstrate such extensive fusion of the metatarsals, and is instead characterized by fusion limited to the proximal end of the element as in Soroavisaurus, Initiornis, and Early Cretaceous enantiornithines (O’Connor, 2009). Because the distal tarsals are fused to the metatarsals in UCMP 139500, body size is not expected to change substantially—this individual clearly does not represent a juvenile. In Early Cretaceous enantiornithines, the proximal tarsals appear to only fuse when the bird reaches skeletal maturity after several years of growth (Hu & O’Connor, 2017). However, increased fusion between the metatarsals and slight increase in size during later ontogeny are both possible.

Table 2 Comparative measurements of North American avisaurid tarsometatarsi (modified from Varricchio & Chiappe, 1995).

Measurement	Mirarce eatoni (UCMP 139500)	Avisaurus archibaldi (UCMP 117600)	Gettyia gloriae (MOR 553E/6.19.91.64)	
Length, MTII	44.1	68.7	28.4	
Length, MTIII	48.2	73.9	30.9	
Length, MTIV	45.9	67.8	28.5	
Width of proximal end	14.5	20.9	9.1	
Width, midshaft	10.9	16.6	∼6.4	
Width of distal end	17.3	24.3	10.7	
Width of trochlea, MTII	7.1	9.9	4.8	
Width of trochlea, MTIII	6.3	8.4	4.0	
Width of trochlea, MTIV	3.1	6.9	∼2.5	
Depth of trochlea, MTII	4.6	7.2	2.9	
Depth of trochlea, MTIII	5.8	9.9	3.9	
Depth of trochlea, MTIV	5.2	8.9	4.3	
Note:

Measurements given in units of mm.

The proximal articular surface of the tarsometatarsus is similar in Mirarce and Avisaurus, which differ from the much smaller Gettyia, thus supporting our reassignment of this species to a new genus. In proximal view the articular surface in UCMP139500 and Avisaurus archibaldi are “bean-shaped” with slight concavities on the midline of the dorsal and plantar margins. A slight concavity is present on the plantar margin in Gettyia and Soroavisaurus but the dorsal margin is convex throughout. In Mirarce and Avisaurus the medial and lateral margins of the proximal articular surface are weakly tapered medially and laterally respectively (defining a blunt 80–90° angle), whereas in Gettyia the medial and lateral margins are rounded.

The tubercle for the attachment of the m. tibialis cranialis is enlarged in all avisaurids but its position on metatarsal II relative to the proximal end varies among taxa. It is located at the midpoint in Mirarce, but more proximally located in Avisaurus and Soroavisaurus and more distally located in Gettyia. Given that small differences in the position of this muscle attachment can have great effect on the moment arm generated by this muscle (O’Connor, Averianov & Zelenkov, 2014), differences in the position of this tubercle suggest that pedal function in the Avisauridae was quite varied, possibly reflecting ecological diversity (Zeffer & Norberg, 2003).

In Mirarce, a very narrow fenestration is present between metatarsals III and IV, although we consider this to be a diagenetic artifact and suggest the metatarsals instead would have articulated tightly along their entire lengths in vivo. Regardless, this is distinct from the fenestrated condition of Soroavisaurus australis caused by lateral curvature of metatarsal IV. This feature, also present in Intiornis, is resolved as a synapomorphy of South American avisaurids. The medial edge of metatarsal II in Mirarce is straight, whereas in Avisaurus there is a curved expansion at its midpoint (the result of a hypertrophied ligamental fossa). This fossa is flatter in Mirarce. The plantar surface of metatarsal II is flat in Gettyia whereas a medial plantar crest is present in Avisaurus and Soroavisaurus; the crest is only weakly developed in Mirarce.

The metatarsal II trochlea is distinctly expanded in Avisaurus and Soroavisaurus such that the medial condyle is medially splayed. In Gettyia the trochlea has a more normal appearance with a flat dorsal margin and subequal condyles. In Mirarce, this condition is intermediate; although the condyles are asymmetrical, it is not developed to the extent in Avisaurus and the trochlea maintains a relatively normal appearance in distal view. A metatarsal II trochlea that is wider than that of metatarsal III is resolved as a synapomorphy of the polytomy that includes Mystiornis and the Avisauridae. The medial condyle of the metatarsal III trochlea is modified in all avisaurids although there is some variation. The intercondylar groove is poorly developed on the dorsal surface of Avisaurus but well-developed and continuous with a small dorsal trochlear depression in Gettyia and Mirarce. In Avisaurus archibaldi the medial condyle projects farther distally and plantarly than the lateral condyle, whereas in Gettyia and Mirarce the two condyles are equal in distal extent and the plantar projection of the medial condyle is less extreme. The avisaurid metatarsal IV trochlea is described as crescent shaped in distal view—this is most apparent in Avisaurus. This morphology is weaker in Mirarce and Gettyia, where the condyle is less mediolaterally compressed. A medially excavated metatarsal IV trochlea is resolved as a synapomorphy of North American avisaurids. In Avisaurus the trochlea is proportionately wider and the distal margin is angled medioproximally—laterodistally, whereas in Mirarce, Gettyia, and Soroavisaurus the distal margin of metatarsal IV trochlea is rounded in dorsal view. In dorsal view this trochlea is expanded medially to contact metatarsal III and distally enclose a vascular foramen in Avisaurus, Gettyia, and Soroavisaurus, a structure, that is, lacking in Mirarce. In Mirarce the proximomedial margin of the metatarsal IV trochlea bears only a very small, medially-directed tubercle in place of the large flange observed in Avisaurus, Gettyia, and Soroavisaurus. These differences strongly support our referral of Gettyia to a new genus, also reinforced through cladistic analysis.

Evolutionary trends

The Enantiornithes have a long evolutionary history, first appearing in the 131 Ma Huajiying Formation in China and surviving up to the K–Pg boundary, documented in the end Maastrichtian Hell Creek Formation. Attempts to understand evolutionary trajectories during the 65 million year (plus) of history of this clade have been obfuscated by the poor resolution of the Late Cretaceous fossil record. Despite the paucity of Late Cretaceous specimens, several trends have been suggested: a general increase in range of body sizes, particularly at the upper limit; a greater degree of fusion of compound elements, and the appearance of advanced flight-related features evolved in parallel to the neornithine lineage (e.g., manual reduction, loss of teeth, increase in size of sternal keel) (Chiappe & Walker, 2002; O’Connor, 2009). As one of the most complete known Late Cretaceous enantiornithines, Mirarce contributes data that significantly expands our understanding of evolutionary trajectories in the Enantiornithes. Mirarce is much larger than any Early Cretaceous enantiornithine (close to turkey-sized), in keeping with observations that some Late Cretaceous taxa achieved much larger body size.

Fusion is absent in the distal metatarsals as in other enantiornithines. This absence of fusion may be related to the protracted skeletal development observed in enantiornithines (Chinsamy, Chiappe & Dodson, 1995). The well-fused proximal carpometacarpus and tarsometatarsus, distal tibiotarsus and fused pelvic girdle observed in Mirarce are also found in some Early Cretaceous specimens (e.g., Qiliania, Concornis), thus no strong trend toward increased fusion during enantiornithine evolution is observed. For example, the distal carpometacarpus remains unfused in Late Cretaceous taxa (Elsornis, Neuquenornis, and Enantiornis) and complete fusion of the tarsometatarsus is not observed in any taxon, despite the fact this element is fully fused even in the earliest Cretaceous ornithuromorphs (Wang et al., 2015b). This cannot be considered a product of preservation or sampling because ornithuromorphs are known from far fewer specimens. Only the skull displays new instances of fusion, with the premaxillae and dentaries rostrally fused in all Late Cretaceous specimens. Although rare, fusion of the premaxillae is present in some Early Cretaceous enantiornithines, but no Early Cretaceous specimen has an extensive mandibular symphysis like that observed in Gobipteryx and neornithines. Overall, however, degree fusion of compound bones remains similar across the clade Enantiornithes regardless of geological age.

Although incomplete, the preserved sternal fragment of Mirarce indicates the presence of a well-developed ventral keel. This preserved morphology is very similar to that observed in Neuquenornis. In contrast, Elsornis reveals a condition similar to that of Early Cretaceous enantiornithines with a poorly developed, cranially forked carina (Chiappe et al., 2007). With the exception of the ratites, all living birds possess a large ventral keel. The poorly developed keel in Elsornis may similarly be related to inferences that this taxon was flightless, or suggests this taxon falls outside the lineage in which an unforked, rostrally extending keel evolved in the Enantiornithes (in parallel to the Ornithuromorpha). The former hypothesis is supported by this phylogenetic analyses (Fig. 18), which places Elsornis in a clade of derived taxa including the Avisauridae (O’Connor & Zhou, 2013). The distribution of advanced sternal morphologies among derived enantiornithines is unknown, currently recognized only within the Avisauridae and may potentially represent a synapomorphy of this lineage. However, we consider it likely that flight-related specializations like, such as a ventrally deep keel has, had a broader distribution within the Enantiornithes and may have evolved multiple times.

Other derived features of the flight apparatus limited to the Avisauridae include a narrow furcula (though also seen in Neuquenornis) and the presence of remige papillae, identified for the first time in Mirarce. In modern birds, the function of these osteological structures is to provide a reinforced attachment surface for the calami of the secondary remiges, and in doing so to transfer aerodynamic forces from the feathers to the wing skeleton (Edington & Miller, 1942; Hieronymus, 2015). Turner, Makovicky & Norell (2007) report a statistically significant correlation between the reduction or loss of flight capabilities and the reduction or absence of ulnar remiges in modern birds, concluding that the prominence of this osteological characteristic is a good proxy for flight capacity. We therefore conclude, based on the remarkable size and prominence of the remige papillae observed in Mirarce (Fig. 9), that this taxon had advanced capacities for powered flight. This further supports hypotheses that at least some lineages of enantiornithines convergently achieved more advanced aerial capabilities by the Late Cretaceous. In contrast, the pygostyle morphology of Mirarce exhibits the typical, proximally-forked and distally-constricted morphology that characterizes nearly all enantiornithines including Halimornis and Parvavis, indicating that this feature was static in all known lineages extending into the Late Cretaceous.

Notably, the pedal phalanges in Mirarce are not markedly elongate as in other enantiornithines. However, compared to Cretaceous members of the crown-ward clade the pedal unguals in Mirarce are strongly curved, a morphology also common in enantiornithines. Differences in hindlimb function are suggested by the numerous scars and rugosities present on the femur, hinting at very different musculature—and possibly also ecological habits—than in Early Cretaceous enantiornithines.

Considering that the Cretaceous evolutionary history of the Enantiornithes is nearly as long as that of Neornithes, there is surprisingly little observable morphological variation through time, though significantly, elements that show the greatest variation in Early Cretaceous avifauna (such as the skull and sternum) are not well known in the Late Cretaceous enantiornithine fossil record. The most extreme morphological diversity resides in the tarsometatarsi from the Lecho Formation described by Chiappe (1993). These data do indeed suggest that Late Cretaceous enantiornithines in this avifauna were ecologically diverse. Attempts to understand evolutionary trends in Enantiornithes remain impeded by the limited Late Cretaceous fossil record, thus the information preserved in the skeleton of Mirarce is all the more critical, paving the way for future significant advances in this area.

Supplemental Information

Supplemental Information 1 New phylogenetic characters & scorings.

Click here for additional data file.

Supplemental Information 2 Phylogenetic analysis data matrix.

Click here for additional data file.

We are grateful to David Strauss for photographs of the specimen, Gregory Arena for the technical drawings of the specimen, Scott Hartman for the customized skeletal reconstruction, Brian Engh for the living reconstruction artwork, and Steve Henriksen for facilitating the commission of Brian’s art. We also thank Peter Kloess for photographing ulnar quill knobs in the UCMP comparative collection, Andrew McDonald for creative assistance with naming the new taxon, Tom Stidham and Nikita Zelenkov for helpful discussions. We thank Mark Goodwin for his role in preparing the specimen. Finally, we are grateful to Nikita Zelenkov, Luis Chiappe, and one anonymous reviewer for their constructive criticism and helpful comments during the formal review process.

Institutional Abbreviations

FMNH Field Museum of Natural History, Chicago, Illinois

MTM Magyar Természettudományi Múzeum, Budapest, Hungary

PVL Fundación Instituto Miguel Lillo, Tucumán, Argentina

RAM Raymond Alf Museum of Paleontology, Claremont, California

UCMP University of California Museum of Paleontology, Berkeley, California.

Additional Information and Declarations

Competing Interests

Author Contributions

Data Availability

New Species Registration

The authors declare that they have no competing interests.

Jessie Atterholt conceived and designed the experiments, analyzed the data, contributed reagents/materials/analysis tools, prepared figures and/or tables, authored or reviewed drafts of the paper, approved the final draft.

J. Howard Hutchison conceived and designed the experiments, approved the final draft, found and prepared the fossil specimen.

Jingmai K. O’Connor conceived and designed the experiments, analyzed the data, contributed reagents/materials/analysis tools, prepared figures and/or tables, authored or reviewed drafts of the paper, approved the final draft.

The following information was supplied regarding data availability:

The material described is stored at University of California Museum of Paleontology (UCMP139500).

The following information was supplied regarding the registration of a newly described species:

Publication LSID: urn:lsid:zoobank.org:pub:269CEBCA-EC05-425D-B71D-1A5507C8E48B;

Genus name Mirarce LSID: urn:lsid:zoobank.org:act:A90E4FD8-999B-4B7D-BE33-B288D10FC8E8;

Species name Mirarce eatoni LSID: urn:lsid:zoobank.org:act:58005BE1-E4F5-4B7C-9A0C-FD94A0B80F30;

Genus name Gettyia LSID: urn:lsid:zoobank.org:act:1CF0E687-904D-44CC-BC26-99C29080D9CD.

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
