# Peer review of "The most complete enantiornithine from North America and a phylogenetic analysis of the Avisauridae"

_PeerJ, doi:10.7717/peerj.5910_

## Round 0.1 · original submission · Major Revisions

· Academic Editor

Major Revisions

Dear Jessie,

I have now received three reviews of your paper submitted to PeerJ. The reviewers recommend a number of improvements, which should be addressed before re-submission.

In particular, you need to deal with the issues regarding your cladistic analysis mentioned by Reviewer 1 and pay more attention to the older literature, as indicated in Reviewer 3's comments.

Please, together with your unmarked revised manuscript, provide a marked-up copy as well as a document explaining how you have addressed each of the points raised by the reviewers.

Thank you for your attention.

Best regards,
Fabien

Reviewer 1 ·

Basic reporting

The article is written in a fluent English, and only a few terminological terms (mostly are typos) need to be corrected: see attached pdf with annotated comments.
Both introduction and referenced literature are appropriate.
The structure of the article conforms to PeerJ format.
The figures are appropriate, at a good resolution and add significant information to the text.

Experimental design

This is an original research.
The research questions are well defined, and it is clearly explained how this study fills previous gaps.
Methods are described, but the described results cannot be replicated (see below).

Validity of the findings

The study describes a new specimen and erect a novel taxon with a significant relevance in the field of Mesozoic vertebrate paleontology.

The results of the phylogenetic analysis cannot be replicated with the data matrix provided in the submission.
The authors wrote that they obtained 196 shortest trees of 560 steps each.
Using their data matrix, and performing 1000 TBR rounds, saving ten trees per replication (not one as in the protocol followed by the authors), I obtained 818 shortest trees of 558 steps (two steps shorter than those reported by the authors). I encourage the authors to perform a more exhaustive search than the protocol described in the manuscript, to be sure that their analysis has adequately sampled the tree islands.
Furthermore, there are two problematic elements in the data matrix that need to be solved:
1- The OTU Flexomornis is scored exclusively for five characters, and in all of them, the score is “0”: thus, that OTU is redundant with a subset of the scores of the dromaeosaurid Outgroup, and results a wildcard with no resolution, that is placed in all internodes of the topology: I suggest that the authors remove that OTU a priori from the data set, following the principle of safe taxonomic reduction.
2- The provided data matrix is not properly formatted. In particular, one taxonomic unit (OTU named “shenqi” in the data matrix, which I assume it corresponds to the taxon Shenqiornis) lacks a character score. It is unclear what score is missing, but as a consequence, TNT automatically use the first letter in the name of the successive OTU (“cathyaor” , which I assume it corresponds to Cathayornis) as missing last score for the OTU “shenqi”.
The absence of a character score in one OTU may have significant impacts on the analysis, as it may result in a badly formed string of character states. I encourage the authors to check carefully the scores in “shenqi” and to correct the missing score. Alternatively, that OTU should be removed as potentially spurious.

Another issue, already noted by the authors, is the overall topology resulted for Enantiornithes. The tree obtained markedly challenges the consensus among the majority of the previous studies on enantiornithines. In particular, the placement of forms like Protopteryx and Iberomesornis as more derived than Upper Cretaceous forms like Gobipteryx and avisaurids is quite unexpected, and raises the doubt that character polarity among the enantiornithines was not properly assessed. This suspect is also reported by the authors who stated that “The absence of basal ornithuromorphs is probably somewhat affecting enantiornithine relationships”. I strongly agree with the last sentence, and suspect that the use of just two very derived ornithuromorphs and the absence of basal members of the direct enantiornithine sister taxon has shaped the bizarre internal relationships resulted among Enantiornithes.
To test the hypothesis that the topology among the enantiornithines is biased by taxon sample among ornithuromorphs, I repeated the analysis omitting a priori the two advanced ornithuromorphs from the taxon sample. The resulted shortest trees placed Protopteryx as basalmost enantiornithine (in agreement with the majority of previous studies), and recovered Enantiophoenix and Mystiornis as very closely related to avisaurids (again, in agreement with previous studies). This test demonstrates that taxon sampling among the ornithuromorphs has a relevant impact on enantiornithine topology.
I encourage the authors to repeat the analysis including a couple of basal ornithuromorphs (e.g., Hongshanornis, Archaeorhynchus, or Archaeornithura) and/or to remove the two derived ornithuromophs, because – as the authors themselves noted – the bizarre topology resulted in their analysis is clearly affected by a poor taxonomic sample among the ornithuromorphs. Ichthyornis and Gallus are too derived for being used as appropriate proxies of the ancestral ornithuromorph morphology.

In summary, the published results of the analysis cannot be replicated, due to a series of problematic issues in both the data matrix and the protocol followed. I suggest that the authors:
- Correct the missing score in Shenqiornis,
- Remove Flexomornis, bacause it is a wildcard that lacks any significant score,
- Rename all OTUs in the matrix file with their proper full names, to avoid misinterpretations of the results,
- Perform more exhaustive analyses (for example, 1000 TBR rounds saving 10 trees per replication),
- Add at least two basal ornithuromorphs to the sample and/or remove the two advanced ornithuromorphs.


A last comment is related to the taxonomy introduced in this study.
The authors erect a new genus for a taxon that results phylogenetically bracketed by two species previously referred to the genus Avisaurus. This means that the authors implicitly consider “Avisaurus” as paraphyletic, or eventually, polyphyletic.
It is interesting that in the previous studies including Avisaurus gloriae and Avisaurus archibaldi, their sister taxon relationships was based exclusively on a single feature, the shape of the distal end of metatarsal IV. In the current study, the authors have demonstrated that this single feature cannot diagnose a clade, and, thus, that the diagnosis of the genus Avisaurus is not valid. This must be more strongly remarked in the text. In particular, the authors should explicitly state if they suggest a new genus name for “A.” gloriae, or, alternatively, if they consider “A.” gloriae to be referred to the same genus of its sister taxon, Mirarce.
Note that in Table 2, the new taxon is labelled “Avisaurus sp.”: please, correct that label to avoid further confusion.
An alternative solution is to emend the diagnosis of the genus Avisaurus, in order to include also the species “Mirarce eatoni”, as Avisaurus eatoni: this alternative solution (which, apparently, was an initial option for the authors, as evidenced by the use of “Avisaurus sp.” in Table 2) has the merit of not revolutioning the taxonomy of A. gloriae relative to A. archibaldi, and thus keeps stable the avisaurid taxonomy used in the last two decades.
Whatever the option preferred by the authors, it must be more explicitly discussed and defined, to avoid future misinterpretations of the term “Avisaurus”.

Additional comments

I have no general comments.

Annotated reviews are not available for download in order to protect the identity of reviewers who chose to remain anonymous.

·

Basic reporting

no comments.

Experimental design

no comments.

Validity of the findings

no comments.

Additional comments

This is well-written paper which describes very nicely preserved late cretaceous north american enantiornithine bird. The paper meets all the criteria of the journal and definitively should be published.
I have a number of rather minor comments.

Line 54. I am not sure that it’s clear what controversy the authors mean. There was a debate on the affinities of the enantiornithes, but apparently not on the identification. Maybe some references (which hare now lacking) would explain this point and clarify this. The discovery of the first articulated skeletons in Spain and China initially had no effect on the acceptance of the Enantiornithes as the clade, because Spanish and Chinese fossils were first described as primitive birds of unclear affinities. It was Larry Martin who united many primitive avians within Enantiornithes in his 1983, 1987 and 1995 (submitted in 1992) papers. His contribution was followed by many others. Luis Chiappe contributed in assigning Avisauridae to the same clade in 1992.
Line 68. The statement about four taxa is not correct. At least, the “Nanantius” valifanovi is known from partial skeleton (Kurochkin, 1996), and more embryos were recently described as Gobipipus (Kurochkin et al 2013).
Lines 152-155. The sequences in which different skeletal elements of the holotype are mentioned is rather unusual. It seems more logical to give them in an anatomical sequence.
Line 179. It will be better to list characters anatomically – first axial skeleton, then forelimb, followed by the hindlimb. The craniocaudal midpoint of the shaft of metatarsal II is not clear. In the pes of bids, usually dorsal and plantar sides are recognized, as well as proximal and distal.
Line 184. “Strongly developed keel” is not a clear statement. Does it refer either to its depth, or craniocaudal extent, or robustness, or all of it in combination? The description on line 245 does not clarify that. It should be specified.
Line 226. Typically enantiornithine pygostyle is not clear, because Pengornithidae have a different one.
Line 268. Given that Pengornithidae have a different proximal humerus, “typical enantiornithine” humerus sounds unclear.
Line 322. Modern birds have differently built carpometacarpi and the morphology of the trochlea carpalis differs in various taxa.
line 510. My former experience shows that 1000 replications may not be enough to find the shortest tree even in a medium-sized dataset. The authors may thus try a larger number of replications to test if more shorter trees may be recovered.
Line 553. Mystiornis was not assigned to Aviasauridae not because of incomplete fusion – as currently stated in the manuscript, bur rater because of complete fusion of metatarsals in Mystiornis – see Kurochkin et al 2011, left column, third line from below. The complete fusion of metatarsals is the character that prevents placing Mystiornis within Enantiornithes but rather unites it with ornithurae. So, the discussion on lines 554-557 should be modified – even the adult Enantiornithes never show a complete fusion of metatarsals throughout its entire length, as in Mystiornis.
Line 628. The morphology of the musculus tibialis cranialis will not affect grasping, as the muscle dorsiflexes the metatarsus. Grasping, in contrast, is produced by plantar flexion by toe flexors.
Lines 670-671. Noguerornis comes from Montsec limestones which are older than 125 MYA (see Selden, Nudds 2012 book), and thus may be the oldest, or at least as old as the oldest Chinese enantiornithines.
Line 727. Although poorly described, a diversity of the late Cretaceous Enantiornithes from Uzbekistan is notable, though represented mainly by coracoids (described in several papers by Panteleyev).

·

Basic reporting

I'm thrilled to see this specimen finally described! It is a very important specimen, one of the most complete and best preserved Late Cretaceous enantiornithine, and the authors have done a good job with the description and the superb photographs.

Experimental design

I think the authors need to be careful with the cladistic analysis--it is plagued with missing data. I know they can't do much about it--we just need new and more complete specimens--but the results should be considered as tentative. This said, I found the recovery of North American and South American subclades of avisaurids very interesting.

The description is adequate but it could be more extensive and better for some bones. The description of the vertebrae is very succinct--there's no mention of the parapophyses, the epipophyses, and other important structures that are clearly visible (even labeled) in the figures. This specimen gives the authors the opportunity to provide a more extensive anatomical description of 3-dimensionally preserved vertebrae that are ofter difficult to interpret in the bidimensional material from China. Likewise, in the description of the coracoid there is no mention of the dorsal fossa, which is evident in the figures, and the fact that the coracoidal nerve foramen opens above (i.e., not inside) of such fossa--this is important given the variability of this character among enantiornithines. I strongly encourage expanding the description of these bones.

Validity of the findings

I don't have much to comment on the overall structure of the paper but see my detail comments in the next section.

Additional comments

1. Between lines 54-56, the authors say that the identification of the El Brete enantiornithines "was initially met with controversy due to the disarticulated and isolated nature of the “El Brete” material, which lasted until the discovery of articulated skeletons from Early Cretaceous deposits in Spain and China." I believe the authors should give credit to the importance of the discovery of the skeleton that would later become the holotype of Neuquenornis. My 1992 paper on avisaurid tarsometatarsi, later cited by the authors, had the goal of documenting the avian and enantiornithine status of Avisauridae. The holotype of Neuquenornis played a key role because it was the first relatively complete specimen interpreted as an enantiornithine (note that this wasn't the case with the initial description of Iberomesornis) that also shared synapomorphies with the isolated bones from El Brete. Therefore, I think the authors should re-write the phrase mentioned above to something like "..until the discovery of articulated specimens from other Cretaceous deposits in Argentina, Spain, and China".

2. Between lines 67-72, the authors say that there are only 4 late Cretaceous taxa represented by articulated specimens: Parvavis, Elsornis, Neuquenornis, and a series of Mongolian embryos. I think this is a bit arbitrary and suggest some editing in order to incorporate other multi-bone discoveries such as Halimornis, Alexornis, and Gobipteryx (originally described as Nanantius by E. Kurochkin).

3. On lines 71-72, the authors refer a series of embryos from Mongolia to Gobipteryx. This has been an unfortunate historical misconception because these embryos, first described by A. Elzanowski, cannot be assigned to Gobipteryx. They can be identified as enantiornithines but assignation to the genus Gobipteryx is not justified. Even Elzanowski recognized that in his original studies. I suggest that the authors don't assign these embryos to Gobipteryx but simply refer to them as 'enantiornithine embryos'.

4. In addition to the Late Cretaceous discoveries mentioned between lines 85-86, the authors should include other remains listed by Chiappe & Walker (2002 - Chapter 11 of the book Mesozoic Birds)--these belong to historical collections by Yale and Princeton.

5. In the vertebral description, the authors use the term post/pre zygOpophysis but the correct anatomical term is zygApophysis (pre/post etc).

6. Lines 250-251, please note that a furrow in the costal surface of the scapula has also been described for the El Brete enantiornithine material (see Chiappe, 1996 - Munchner Geowiss. Abh. 30: 203-244; and Chiappe and Walker, 2002 in Mesozoic Birds).

7. It would be helpful to indicate some of the characters of the omal end of the coracoid in the photographs.

8. Line 263 has two 'the'.

9. Line 268-269, the authors say that the for the humerus, 'the proximal end is typically enantiornithine in profile: concave centrally rising dorsally and ventrally (Fig. 8)". I agree but it would be helpful for the reader to have general references to humeri such as Chiappe (1996 - ) and Chiappe and Walker (2002)--these papers describe the feature mentioned above and illustrate a variety of humeri.

10. Likewise, in line 271, please cite Chiappe (1996 - Memoirs of the Queensland Museum 39: 533-554) for Enantiornis. This paper provides a description and illustrations of the holotype of Enantiornis leali, which were not provided by Walker in his 1981 paper.

11. In line 279-280, the authors say (in parentheses) that the pneumotricipital fossa of the humerus "is perforated only in PVL 4022 [Walker & Dyke 2009]". For the record, Chiappe and Walker (2002) mentioned the presence of a pneumotricipital foramen in the humerus of enantiornithine specimen PVL 4022 several years prior to Walker & Dyke's publication (it was also figured). Here is the exact text used in Chiappe and Walker (2002): "Although the humerus of several euenantiornithines bears a distinct pneumotricipital fossa, a pneumotricipital foramen is known only for an isolated humerus from El Brete (PVL-4022; Fig. 11.8.C)."

12. In regards to the pelvis, the authors may want to make comparisons with the specimens featured in Chiappe & Walker (2002). I can't see it well in the figures but the position of the antitrochanter is important--what do they authors mean by a "dorsal antitrochanter"? Does it mean that it is positioned in the ldorso-dorsocaudal corner of the acetabulum? Labeling the figure of the pelvis would be important as well. The description could be revised, there is at least one type (lease instead of least).

13. Line 365. What do the authors mean by "posterior trochanteric crest"? This is the posterior trochanter that they are referring to. Calling this a posterior trochanteric crest is confusing: it makes it look as if they are establishing a homology with the trochanteric crest of early theropods (e.g., ceratosaurians).

14. Lines 659-664. I'm not sure I would describe the xiphoid process of the new taxon as V-shaped; it looks pretty straight and in this respect not that different from what is visible in some other enantiornithines. Also, I would be careful about comparing this with Neuquenornis because the caudal sternum of the latter is largely missing. Likewise, the hypocleidium of Neuquenornis is incomplete, so I'm not sure you could determine its length.

---

## Round 0.2 · Minor Revisions

· Academic Editor

Minor Revisions

Dear Jessie,

Thank you very much for addressing the comments of the reviewers of the first round. Two of these reviewers examined the second version of your work and recommended minor additional revisions. These include registering the new genus name in Zoobank and carefully editing your text (removing notes such as "add citations", "WHO?" etc and correcting misspellings such as "asand", "Phalacocorax" etc.).

Best regards,
Fabien

Reviewer 1 ·

Basic reporting

No particular comments compared to the previous submission: the authors have addressed my comments and suggestions.
I only note that some notes and comments - that clearly belong to the in progress draft and apparently were not meant to be included in the submission file - are still present along the text (e.g., lines 306-7). The authors are invited to carefully check the text and remove any comment or note of the draft present in the text file before uploading the final submission.

Experimental design

The authors have followed all suggestions by the reviewers on both description and phylogenetic analysis.

Validity of the findings

As expected, updating the taxon sample of Ornithuromorpha the relationships among the enantiornithines substantially agree with most of previous studies.
(Although I usually do not evaluate a novel phylogenetic scenario based just on a priori expectations taken from previous literature, the re-analysis has confirmed that the topology in the first submission was clearly biased by taxon sampling, and had to be considered with concerns).
The authors have expanded the comparative and descriptive section, which is welcome given the relative paucity of Late Cretaceous enantiornithines with a comparable preservation as the new taxon.

Additional comments

A few taxonomic remarks. Although they may seem just formal notes, they prevent the manuscript from being emended by future publications on enantiornithines:

The new genus name for "A." gloriae should be registered in ZooBank, otherwise it is not valid under the ICZN: the ZooBank code of Gettyia must be included in the manuscript.

In line 156-7, the proper definition of a node-based clades Avisauridae should be "the LAST common ancestor of Neuquenornis volans and Avisaurus archibaldi plus all its descendants (Chiappe 1993)."
The "last" adjective must be added, otherwise, the clade definition is ambiguous (e.g., even the first vertebrate is a common ancestor of Neuquenornis and Avisaurus: you should explicitly state that Avisauridae refers to the least inclusive among all clades including the two mentioned species).

In line 705, the correct form should be: "GETTYIA gloriae (Varricchio & Chiappe, 1995) new comb.", because the authors have not erected a new species name: they have just erected a new combination for the previously-erected species "gloriae" with a new genus name.

·

Basic reporting

as stated in my previous review.

Experimental design

as stated in my previous review.

Validity of the findings

as stated in my previous review.

Additional comments

The new version of the manuscript may be published after some minor corrections. The manuscript contains many typos and mispellings, especially in the newly added or edited parts (e.g. lines 64, 89 et cet.), and require careful edition. Some of the added references are not on the reference list. The figure 17 looks unfinished.

---

## Round 0.3 · accepted · Accept

· Academic Editor

Accept

Dear Jessie,

I'm pleased to accept your manuscript for publication in PeerJ.

Please, make sure that only the publication LSID appears in the body of the article (in the Methods section). [The three other LSIDs, corresponding to the two new genera and the new species will be at the end of the article, in the Additional Information section]

Also make sure that the background of every plate is homogenously 100% black in the final version. Currently, it does not seem to be the case for (at least) Fig. 2, 10, 16, and 17.

Best regards,
Fabien

#